# Clonal evolution in myelodysplastic syndromes

Pedro da Silva-Coelho[1,2,*], Leonie I. Kroeze[1,*], Kenichi Yoshida[3,*], Theresia N. Koorenhof-Scheele[1], Ruth Knops[1], Louis T. van de Locht[1], Aniek O. de Graaf[1], Marion Massop[1], Sarah Sandmann[4], Martin Dugas[4], Marian J. Stevens-Kroef[5], Jaroslav Cermak[6], Yuichi Shiraishi[7], Kenichi Chiba[7], Hiroko Tanaka[7], Satoru Miyano[7], Theo de Witte[8], Nicole M.A. Blijlevens[9], Petra Muus[9], Gerwin Huls[9,10], Bert A. van der Reijden[1], Seishi Ogawa[3] & Joop H. Jansen[1]

Cancer development is a dynamic process during which the successive accumulation of mutations results in cells with increasingly malignant characteristics. Here, we show the clonal evolution pattern in myelodysplastic syndrome (MDS) patients receiving supportive care, with or without lenalidomide (follow-up 2.5–11 years). Whole-exome and targeted deep sequencing at multiple time points during the disease course reveals that both linear and branched evolutionary patterns occur with and without disease-modifying treatment. The application of disease-modifying therapy may create an evolutionary bottleneck after which more complex MDS, but also unrelated clones of haematopoietic cells, may emerge. In addition, subclones that acquired an additional mutation associated with treatment resistance (TP53) or disease progression (NRAS, KRAS) may be detected months before clinical changes become apparent. Monitoring the genetic landscape during the disease may help to guide treatment decisions.

[1] Laboratory of Hematology, Radboud University Medical Center, Geert Grooteplein Zuid 8, 6525 GA Nijmegen, The Netherlands. [2] Department of Haematology, Centro Hospitalar de São João and Faculdade de Medicina da Universidade do Porto, Alameda Professor Hernâni Monteiro, Porto 4200-319, Portugal. [3] Department of Pathology and Tumor Biology, Graduate School of Medicine, Kyoto University, Yoshida-Konoe-cho, Sakyo-ku, Kyoto-shi, Kyoto 606-8501, Japan. [4] Institute of Medical Informatics, University of Münster, Albert-Schweitzer-Campus 1, 48149 Münster, Germany. [5] Department of Human Genetics, Radboud University Medical Center, Geert Grooteplein Zuid 8, 6525 GA Nijmegen, The Netherlands. [6] Institute of Hematology and Blood Transfusion, U Nemocnice 1, 128 20 Prague 2, Czech Republic. [7] Human Genome Center, Institute of Medical Science, The University of Tokyo, 4-6-1, Shirokanedai, Minato-ku, Tokyo 108-8639 Japan. [8] Department of Tumor Immunology, Radboud University Medical Center, Radboud Institute for Molecular Life Sciences, Geert Grooteplein Zuid 8, 6525 GA Nijmegen, The Netherlands. [9] Department of Hematology, Radboud University Medical Center, Geert Grooteplein Zuid 8, 6525 GA Nijmegen, The Netherlands. [10] Department of Hematology, University Medical Centre Groningen, PO Box 30001, 9700 RB Groningen, The Netherlands. * These authors contributed equally to this work. Correspondence and requests for materials should be addressed to S.O. (email: sogawa-tky@umin.ac.jp) or to J.H.J. (email: Joop.Jansen@Radboudumc.nl).

Myelodysplastic syndromes (MDSs) are a heterogeneous group of haematopoietic neoplasms characterized by abnormal differentiation, dysplasia and peripheral blood cytopenias. Progression towards acute myeloid leukaemia (AML) occurs in ∼30% of the patients. Various genetic mutations underlying the pathogenesis of MDS have been identified. Most of the recurrently affected genes can be classified as transcription factors, signal transduction proteins, epigenetic modifiers, proteins involved in RNA splicing and proteins of the cohesin complex[1–3]. Typically, in a given MDS patient, several mutations are present simultaneously. Various genes are recurrently mutated in different individuals with MDS and likely play a role in the pathogenesis of the disease (driver mutations), but also random, nonpathogenic mutations that are acquired in individual cells during life are found, as these are clonally expanded together with the pathogenic mutations during the development of the disease (passenger mutations)[4]. Oncogenesis is thought to be a multistep evolutionary process. The successive acquisition of several mutations that confer a selective advantage may result in the emergence of populations of cells that harbour the same set of mutations[5,6].

Both linear and branching patterns of evolution have been described. Linear evolution is characterized by the successive appearance of dominant clones that overgrow their ancestral clone after the acquisition of additional mutations. Branching evolution is characterized by the emergence of different subclones from one common ancestral clone, leading to the coexistence of related (sub)clones that contain a partially overlapping set of mutations[7,8]. The genetic diversity amongst these coexisting subclones may result in a more difficult to treat type of disease, as some of the subclones may be resistant to specific types of therapy.

Several studies have documented the genetic evolution in MDS and AML[5,9–14]. Evolutionary patterns in MDS patients before or without leukaemic transformation are, however, scarce and are often based on the analysis of a limited number of samples per patient. In this study, we performed an in-depth analysis of clonal evolution in MDS patients who were followed over a prolonged period of time. We show that both linear and branched evolutionary patterns occur in MDS, and that clonal evolution can be influenced by treatment.

## Results

**Genetic analysis of MDS patients.** We assessed clonal evolution by whole-exome sequencing (WES) followed by targeted deep sequencing in 11 MDS patients (Table 1). T-cell DNA was used as germline control. In addition, DNA from cultured mesenchymal stromal cells (MSCs) was used as reference in five patients. Six patients received supportive care (transfusions, growth factors) only, whereas five patients also received lenalidomide. To capture all mutations, WES was performed at the first and last as well as at several intermediate time points ($n = 45$). In addition, FLT3-ITD was detected by fragment length analysis. Furthermore, in specific cases, amplicon-based deep sequencing was used targeting a panel of genes recurrently mutated in myeloid malignancies (Supplementary Tables 1 and 2). All identified mutations were validated and quantified by targeted deep sequencing in all available samples of each patient (on average 10,616 fold coverage). In 158 different genes, 176 different acquired somatic mutations were identified (Supplementary Data 1). The median number of acquired gene mutations was 17 (range 8–27) per patient. Of these, a median of four mutations per patient (range 0–6) were present in genes that have previously been implicated in myeloid malignancies and are considered to be driver mutations (Fig. 1a,c). The total number of genetic defects detected in the first sample of each patient correlated with the age of the patient ($P = 0.03$, Fig. 1b), in line with the accumulation of genetic alterations during ageing. The most frequent alterations were nonsynonymous single-nucleotide variants (SNVs) ($n = 145$, 82%) (Fig. 1d). Of all SNVs, 65% ($n = 105$) were transitions, predominantly G:C→A:T (53%, Fig. 1e). Some mutations were detected in all samples from a given patient, whereas others were only seen at early or late time points, indicating genetic evolution (Supplementary Fig. 1). No major influence of therapy on the type of SNVs (transitions or transversions) was observed when comparing early with late mutations in the two different treatment groups (Supplementary Fig. 2). Based on the variant allele frequencies (VAFs) at all available time points (Supplementary Figs 3 and 4), mutations were clustered and clonal composition and evolution patterns were reconstructed (Figs 2 and 3). Results from high-density single-nucleotide polymorphism (SNP) arrays (Supplementary Table 3) and conventional cytogenetic analysis (Supplementary Data 2) were taken along when reconstructing the clonal evolution.

**Clonal evolution in patients treated with supportive care.** Six patients were treated with supportive care only, consisting of transfusions and growth factors (erythropoiesis-stimulating agents, granulocyte colony-stimulating factor and thrombopoietin receptor agonist). In one of these patients (UPN04), just one clone of MDS cells was observed, carrying 12 mutations including 3 mutations in recurrently mutated genes: one ZRSR2 mutation and two different mutations in TET2 (Supplementary Data 1 and Supplementary Fig. 3). The set of mutations carried by this clone remained unchanged over the entire observational period of 8 years, during which the patient's clinical condition remained stable (Fig. 2a).

Two patients (UPN06 and UPN11) showed a linear evolution pattern, in which successive clones, carrying increasing numbers of mutations, overgrew their ancestral clones (Fig. 2b,c). In both cases, concomitant with the emergence and expansion of a clone harbouring a mutation in NRAS, the patient developed leukocytosis (both $>100 \times 10^9$/l, for UPN06 after the last time point) and progression of disease: UPN11 progressed from RCMD (refractory cytopenia with multilineage dysplasia) to RAEB-1 (refractory anaemia with excess blasts-1) and ultimately developed secondary AML (sAML) (Fig. 2b), whereas UPN06 progressed from RARS (refractory anaemia with ringed sideroblasts) towards RAEB-2 (Fig. 2c).

The other three patients who did not receive disease-modifying treatment showed more complex, branching clonal evolution patterns. In UPN03 (Fig. 2d), two divergent subclones emerged from a common ancestral clone. Despite the genetic evolution, the clinical condition of the patient did not evolve significantly over the 8 years of follow-up. Eventually, this patient died of prostate cancer. The other two patients with branching evolutionary patterns progressed towards sAML (Fig. 2e,f). In both patients, mutations in RAS pathway members were observed. In patient UPN05, a KRAS-mutated clone emerged. In patient UPN07, two subclones, one carrying an NRAS mutation and one carrying an RRAS mutation, were derived from a common ancestral clone. The NRAS-mutated clone was dominant at the time of first sampling. Over time, this clone was gradually outcompeted by subclones of the RRAS-mutated clone, with concomitant progression to sAML.

**Clonal evolution in patients treated with lenalidomide.** Five patients who received lenalidomide were analysed, four of whom carried a deletion on chromosome 5q (Fig. 3). All four 5q − patients responded well to lenalidomide (Fig. 3a–d), resulting

**Table 1 | Baseline patient characteristics.**

| UPN | Sex | Age | Duration of AHD (months) | AHD type | MDS subtype (FAB) | MDS subtype (WHO) | IPSS-R | Transformation to AML | Cause of death | Cytogenetic abnormalities | Follow up time (years) | Sampling moments (n) |
|---|---|---|---|---|---|---|---|---|---|---|---|---|
| 1 | F | 51 | 7 | Anaemia | RAEB | RAEB-1 | High | No | NA | del5q, t(X;16) | 11.2 | 19 |
| 2 | M | 62 | 7 | Anaemia | RARS | RARS | Very low | No | TBC | NN | 5.0 | 5 |
| 3 | M | 56 | 40 | Anaemia | RARS | RARS | Very low | No | Prostate cancer | NN | 7.7 | 6 |
| 4 | M | 66 | 43 | Granulocytopenia | RAEB | RAEB-1 | Low | No | NA | NN | 8.1 | 9 |
| 5 | M | 64 | 66 | Thrombocytopenia | RA | RCMD | Int | Yes | AML | +8 | 7.0 | 6 |
| 6 | M | 58 | 6 | Anaemia | RARS | RCMD | Low | No | MDS/pneumonia | +21 | 5.3 | 5 |
| 7 | M | 67 | 81 | Pancytopenia | RAEB | RAEB-1 | Int | Yes | AML | NN | 3.7 | 7 |
| 8 | F | 67 | 40 | Anaemia, Granulocytopenia | RAEB | RAEB-1 | Int | No | NA | del5q, t(1;10) | 11.3 | 13 |
| 9 | F | 73 | 69 | Anaemia | RA | RA | Low | No | Heart failure | del5q, del9q | 6.6 | 13 |
| 10 | F | 57 | 70 | Thrombocytosis | RA | RCMD | Int | No | NA | del5q, del13q | 9.3 | 31 |
| 11 | M | 67 | 0 | NA | RA | RCMD | Int | Yes | AML | NN | 2.4 | 6 |

AHD, antecedent haematological disease; AML, acute myeloid leukaemia; F, female; FAB, French–American–British classification system; Int, intermediate; IPSS-R, Revised International Prognostic Scoring System; M, male; NA, not applicable; NN, normal karyotype. RA, refractory anaemia; RAEB-1, refractory anaemia with excess blasts-1; RARS, refractory anaemia with ringed sideroblasts; RCMD, refractory cytopenia with multilineage dysplasia; TBC, tuberculosis; UPN, unique patient number; WHO, World Health Organization classification system.

in morphological and cytogenetic complete remission. However, when considering the total set of somatically acquired mutations, these four patients showed substantial differences with regard to their clonal evolution patterns. Patient UPN01 (Fig. 3a) initially showed a very good response to lenalidomide, and the MDS clone was reduced to 2% of the bone marrow (BM) population. This response was gradually lost during lenalidomide treatment, as a descendant of the original clone carrying additional heterozygous *RELN* and *TP53* mutations slowly expanded, accompanied by a gradual decline in haemoglobin levels. *TP53* mutations are known to be associated with lenalidomide resistance[15].

In the other three 5q− patients, distinct, nonrelated clonal populations grew out during complete remission. These emerging clones were already detectable at low levels before treatment (Fig. 3 and Supplementary Data 3). In UPN08 and UPN09 (Fig. 3b,c), the MDS clones that dominated haematopoiesis before the start of lenalidomide treatment diminished under treatment to 0.2% and 2% of the bone marrow population, respectively (Supplementary Data 2 and Supplementary Figs 3 and 4). In both patients, however, genetically distinct clones emerged. To confirm that these expanding clones did not harbour any mutations that were found in the previously detected dominant clones, we performed colony assays (CFU-GEMM (colony-forming unit–granulocyte, erythrocyte, monocyte and megakaryocyte)), followed by sequencing of individually picked colonies. This showed that the rising clones did not harbour any of the mutations that were present previously (Fig. 4). In both patients, del(5q)-containing clones were strongly suppressed by lenalidomide, but not completely eradicated. For example, in UPN08, lenalidomide appeared to suppress all the clones present before the start of lenalidomide treatment (containing, among others, a *CSNK1A1* mutation), but mutations remained detectable in ~0.4% of the cells during treatment (Supplementary Fig. 5).

In addition, in the remaining 5q− patient, UPN10 (Fig. 3d), clonal populations were still detectable during complete remission. Under lenalidomide treatment, cells carrying 5q− and 8 other mutations with or without a monosomy 7 (Fig. 3d, red and dark green clones) were strongly suppressed, but a non-5q-deleted ancestral clone (dark blue clone, Fig. 3d) containing 6 mutations remained present. Under lenalidomide treatment, subclones derived from this ancestral MDS clone expanded over time. In addition, a *JAK2* V617F containing clone expanded under lenalidomide treatment (Fig. 3d and Supplementary Fig. 4). Sequencing of single-cell-derived colonies showed that cells

harbouring this *JAK2* mutation did never harbour mutations present in the other subclones, indicating that the *JAK2*-mutated cells represented a separate, unrelated clone (Fig. 4). After 4.5 years of treatment, the patient lost response to lenalidomide: the haemoglobin levels gradually declined and the 5q− clone, which had been suppressed under lenalidomide, slowly expanded. Because of clinical disease progression, lenalidomide treatment was stopped and the patient underwent an allogeneic stem cell transplantation. As a result, MDS cells were undetectable by cytogenetic and fluorescence *in situ* hybridization (FISH) analysis for more than a year, although some patient-derived blood cells could still be detected by quantitative donor–recipient chimerism analysis (<1%, Supplementary Fig. 6). At 19 months after transplantation, a clinical relapse was diagnosed in this patient, with reappearance of the del(5q)-containing clone. Targeted sequencing of a panel of 72 MDS driver genes revealed no additional mutations at the time of relapse. However, 39 months after transplantation, the MDS progressed to RAEB-1 and additional karyotypic abnormalities and a *CUX1* mutation were observed. Upon relapse, the patient was treated with 5-azacitidine for 8 months that led to a reduction in clone size (Fig. 3d) accompanied by an improvement of haemoglobin levels.

The patient without a del(5q) (UPN02, Fig. 3e) received lenalidomide for 16 months and had stable disease. After discontinuation of lenalidomide treatment, the patient received 5-azacitidine for 1 year, resulting in a transient reduction in transfusion frequency. Under this treatment a subclone containing several mutations, including a mutation in *EZH2*, expanded at the expense of a subclone that was dominant before start of 5-azacitidine treatment (containing an *SF3B1* and *CUX1* mutation). Interestingly, after 5-azacitidine treatment was stopped, the *EZH2*-mutated clone disappeared, with concomitant reexpansion of the *SF3B1*- and *CUX1*-mutated clone.

**Clonal composition in different PB and BM cell fractions.** In MDS, the generation of mature blood cells from BM stem and progenitor cells is disturbed, but not completely abrogated. In theory, different mutations might occur in BM cells at different stages of maturation. In addition, specific mutations might block maturation at a particular stage, whereas others might allow maturation up to completely mature blood cells. As a result, diverse mutational landscapes may be observed in cells of different progenitor cell fractions and maturation stages within a

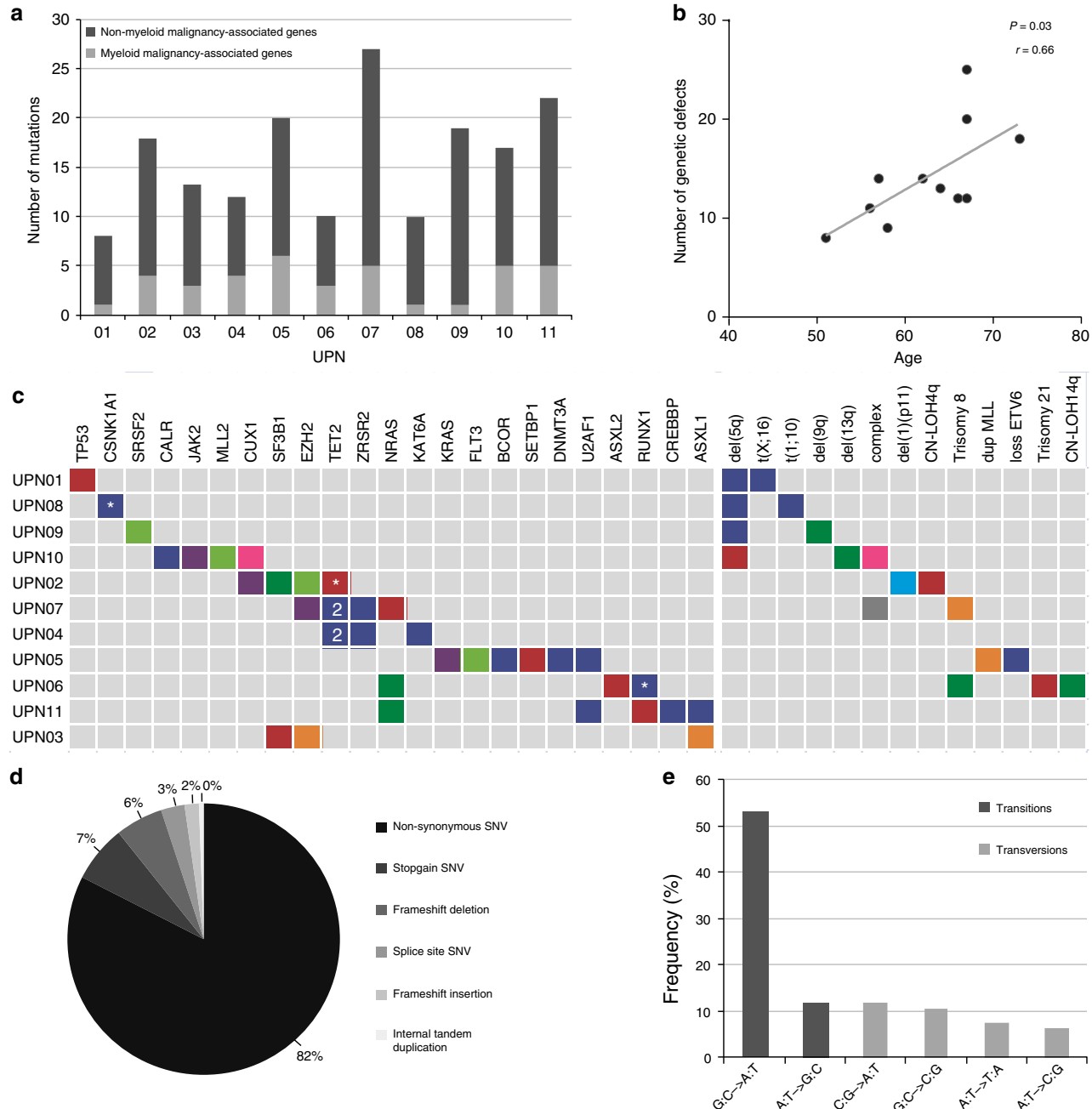

**Figure 1 | Genetic defects in 11 MDS patients. (a)** Number of acquired mutations in 11 patients with MDS, as determined by whole-exome sequencing at several time points (Supplementary Data 1) and confirmed by amplicon-based deep sequencing. In light grey, the number of mutations in genes previously implicated in the pathogenesis of myeloid malignancies are indicated (driver mutations)[2,3,25,41-44], and in dark grey the number of mutations not previously implicated in myeloid malignancies (putative passenger mutations). **(b)** A positive correlation could be observed between age and the number of genetic defects (genetic and cytogenetic defects) at the time of first sampling. Pearson's correlation coefficient (including a two-tailed $P$ value calculated by Student's $t$-test) was determined. **(c)** For each patient, all mutations in genes known to be recurrently mutated in myeloid malignancies are depicted as well as all cytogenetic defects detected by high-resolution SNP array and/or karyotype analysis. The colours match with the (sub)clones as depicted in Figs 2 and 3. *Indicates a mutated gene that is also affected by a copy number gain or loss or by a copy-neutral loss of heterozygosity (CN-LOH); '2' indicates two different mutations affecting the same gene. **(d)** Distribution of the different types of alterations detected in the total set of patients. **(e)** Different types of single-nucleotide changes detected in all patients, with transitions in dark grey and transversions in light grey.

patient. To study this, we isolated DNA from various BM stem (haematopoietic stem cells (HSCs)) and progenitor fractions (common myeloid progenitor, granulocyte–macrophage progenitor and megakaryocyte–erythroid progenitor) of six patients (UPN01, 03, 04, 05, 06 and 10) at several time points in the course of their disease. All mutations detected in the bulk of cells were also detected in all analysed stem and progenitor fractions,

although sometimes with a somewhat different VAF in the various cell fractions (Fig. 5 and Supplementary Figs 7–9). In addition, mutations that arose later during the course of the disease, being characteristic for developing subclones, were present in all stem and progenitor cell fractions at roughly equal frequencies. This suggests that both the early and late mutations arose in early HSCs that are still capable of differentiation into

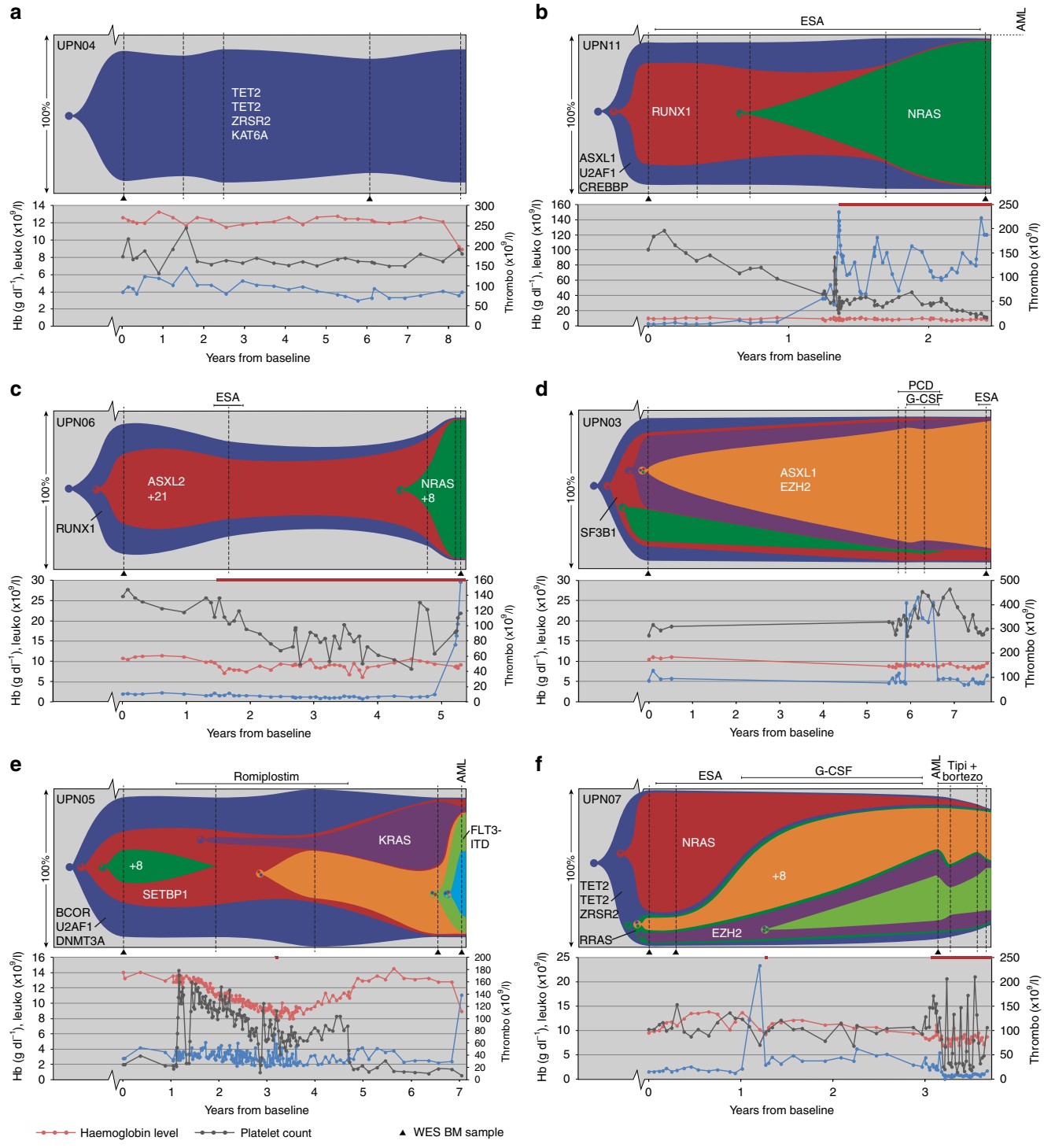

**Figure 2 | Clonal evolution patterns in the bone marrow of MDS patients who received supportive care only.** (**a**) Patient with one single MDS clone without clonal evolution during the 8 years of follow-up. (**b,c**) Two patients showing linear clonal evolution. In both cases, a heterozygous *NRAS* mutation was acquired (green clones), associated with increased leukocyte levels and disease progression. (**d–f**) Patients with a more complex branching clonal evolution pattern. Vertical dashed lines indicate the investigated sampling moments. The samples indicated with a triangle were analysed by WES. Subsequently, all samples were analysed with targeted deep sequencing. Only important genetic aberrations are indicated; a full list of genetic aberrations can be found in Supplementary Figs 3 and 4, Supplementary Table 3 and Supplementary Data 1 and 2. PCD, pentoxifylline, ciprofloxacin and dexamethasone; tipi + bortezo, tipifarnib and bortezomib.

different myeloid lineages. In addition, the mutational burdens in BM and peripheral blood (PB) samples were quite comparable (Supplementary Figs 10–16). In general, the VAFs were somewhat lower in PB, likely caused by a higher percentage of lymphoid cells. The PB granulocyte fraction exhibited comparable mutational burdens to BM samples, indicating that mutated and nonmutated myeloid progenitor cells had a similar capacity to form mature granulocytes.

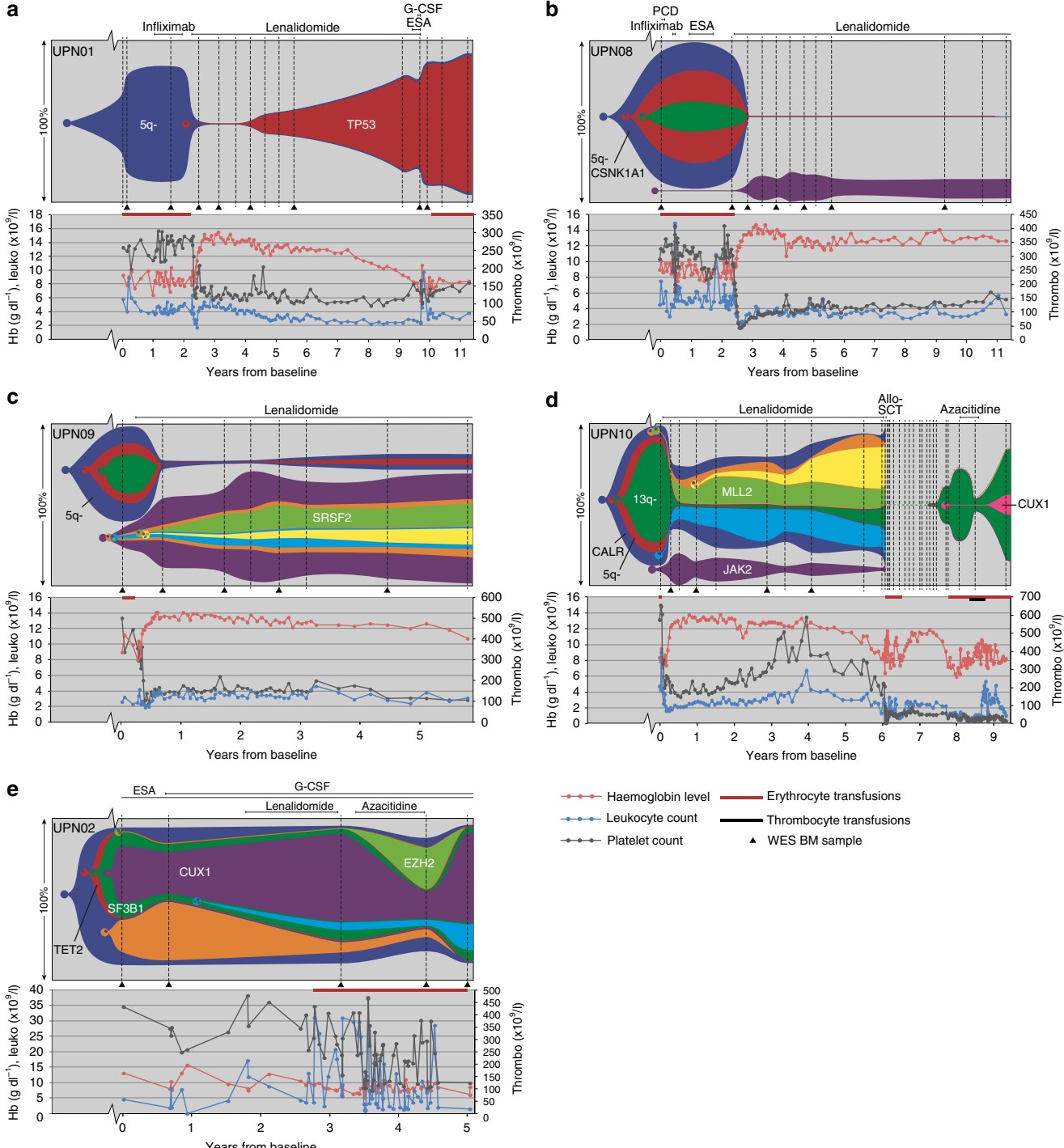

**Figure 3 | Clonal evolution patterns in the bone marrow of MDS patients who were treated with lenalidomide.** (**a**–**d**) Four patients harbouring a del(5q) who responded well to lenalidomide treatment. UPN01 (**a**) shows a linear evolution pattern. In UPN08, 09 and 10 (**b**–**d**), non-MDS-related clonal populations increased in frequency under lenalidomide treatment. The MDS clonal populations followed a linear evolution in UPN08 and 09, and a branched evolution in UPN10. (**e**) Patient with a normal karyotype and without a major response to lenalidomide treatment. This patient shows a branching evolutionary pattern, with a change in clonal composition under 5-azacitidine treatment. Vertical dashed lines indicate the investigated sampling moments. The samples indicated with a triangle were analysed by WES. Subsequently, all samples were analysed with targeted deep sequencing. Only important genetic aberrations are indicated; a full list of genetic aberrations can be found in Supplementary Figs 3 and 4, Supplementary Table 3 and Supplementary Data 1 and 2. PCD, pentoxifylline, ciprofloxacin and dexamethasone.

## Discussion

We studied the mutational spectrum and clonal evolution in MDS patients receiving supportive care, as well as in patients who were treated with lenalidomide. Several patterns of clonal evolution were observed ranging from a patient with a single clone remaining stable for many years to patients with highly dynamic shifts in clonal composition. We confirmed that therapy may influence clonal evolution and that MDS-unrelated clones can arise under treatment[14]. Clonal evolution was observed in both patients treated with lenalidomide and patients treated with

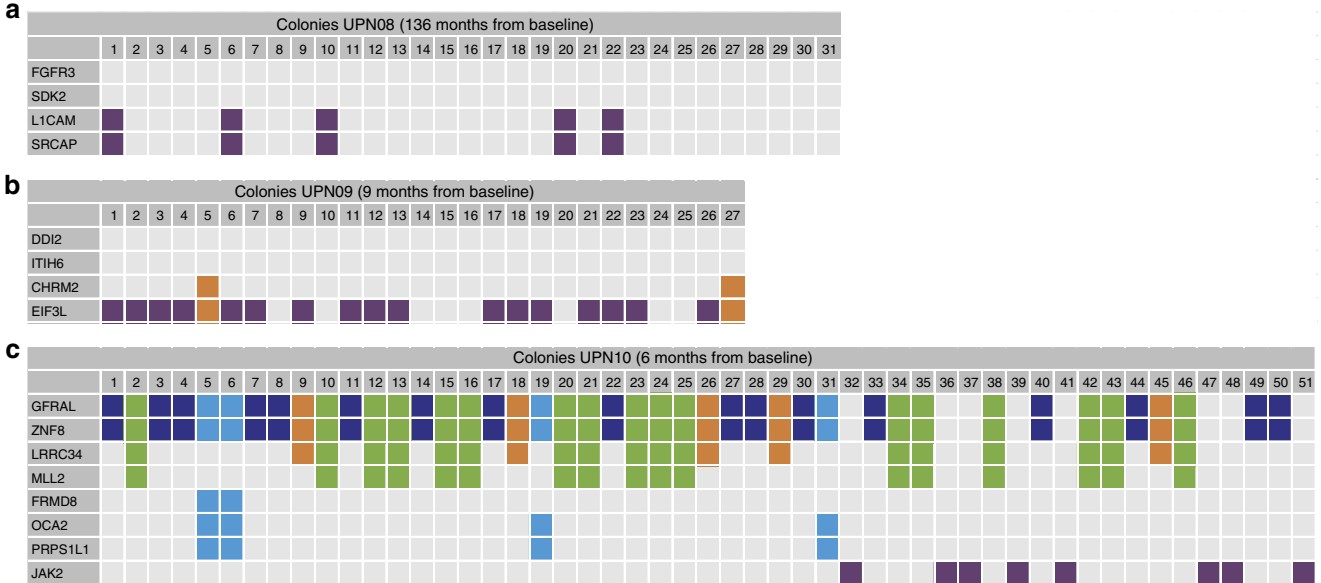

**Figure 4 | Sequencing of single-cell-derived colonies.** To determine which mutations are present together in a single cell and to confirm that cells from the unrelated clones do not harbour any of the ancestral mutations present in the MDS clone, we performed sequencing on single-cell-derived CFU-GEMM colonies. Representative mutations are sequenced from each (sub)clone. (**a**) UPN08: only colonies harbouring the two mutations linked to the unrelated clone are found at this time point. The two investigated mutations from the MDS clone are absent in these colonies. (**b**) UPN09: most colonies only contain an *EIF3L* mutation corresponding to the major unrelated clone. Two colonies harbour an additional *CHRM2* mutation corresponding to a descendent of the major unrelated clone. The mutations from the MDS clone are absent in these colonies. (**c**) UPN10: the *JAK2* clone is an independent clone not containing mutations from the major MDS clone. Furthermore, this analysis confirms that *LRRC34* is a descendent of the major MDS clone that later also acquired an *MLL2* mutation. The mutations in *FRMD8*, *OCA2* and *PRPS1L1* never co-occur with the *LRRC34* and *MLL2* mutations, indicating that these are separate clones. The *FRMD8* mutation appears to be a later event than the acquisition of *OCA2* and *PRPS1L1*. The absence of a mutation (VAF <5%) is indicated in grey. The presence of a mutation (VAF >40%) is indicated with a colour that corresponds to the clones in Fig. 3.

supportive care. Many patients in the supportive care group received growth factors to stimulate haematopoiesis that might have influenced the evolutionary pattern, but since we analysed only a limited number of patients, we cannot draw any conclusions. Three of the patients treated with growth factors eventually progressed to sAML (UPN05, 07 and 11). In all three patients, the clones that ultimately developed into sAML contained a heterozygous mutation in one of the RAS family members. Patient UPN05 and UPN11 acquired a mutation in *NRAS* and *KRAS* respectively, that could be detected months before sAML was diagnosed. In UPN07, two members of the RAS pathway (*NRAS* and *RRAS*) were mutated in separate subclones. *RRAS* mutations are not frequently found in haematological malignancies, but some cases have been described[16]. One patient with juvenile myelomonocytic leukaemia was reported who also carried an *NRAS* and an *RRAS* mutation in separate clones, whereas after chemotherapy only the *RRAS*-mutated clone remained. In UPN07, the initial major clone containing an *NRAS* mutation was outcompeted by the *RRAS*-mutated clone over time. During this shift in clonal composition, no therapy other than erythropoiesis-stimulating agent was given. Previous reports have implicated *RAS* mutations in enhancement of proliferation and progression towards sAML[17–20]. Together with our data, this may indicate that screening for mutations in RAS family members is warranted in MDS, as acquisition of these mutations seems to correlate with the development of more aggressive clones that eventually may result in progression towards sAML. Ultimately, patients who acquire *RAS* mutations might be candidates for specific forms of treatment that target the RAS pathway or its downstream signalling partners, like MEK inhibitors[21].

The mechanism behind the beneficial effect of lenalidomide in patients harbouring a 5q deletion has recently been described[22].

Lenalidomide stimulates the degradation of CSNK1A1 that leads to apoptosis. MDS cells harbouring a deletion of 5q have only one remaining *CSNK1A1* allele, and are therefore thought to be more sensitive to lenalidomide. Many patients eventually develop resistance to treatment that is often accompanied by the acquisition of *TP53* mutations[15,23]. Patient UPN01 initially showed an excellent clinical and molecular response to lenalidomide, but gradually a subclone expanded that had acquired a mutation in *TP53*. This mutation could not be detected before treatment with lenalidomide (at a threshold of 0.2%). The increment of the *TP53*-mutated cells under lenalidomide took considerable time, but eventually the patient experienced recurrence of clinical symptoms. We can only speculate whether intermittent treatment with lenalidomide might have been more beneficial than continuous treatment (sufficient enough to suppress the original *TP53*-negative clone, while stalling the selection of the *TP53*-positive cells), or detrimental (allowing both the *TP53* negative and positive MDS cells to grow). In case of the first possibility, lenalidomide sensitivity might have been preserved over a longer period of time, but to address this, future clinical testing would be required.

UPN08 harboured a mutation in *CSNK1A1* that is described to be mutated in 5–18% of patients with a 5q deletion[24–26]. Although the exact biological role of these mutations is still under investigation, reports so far show a trend towards a decreased response to lenalidomide and a decreased overall survival compared with *CSNK1A1* wild-type 5q− patients[24,25]. In contrast, UPN08 showed a very good response to lenalidomide with a clinical and cytogenetic complete remission maintained for already more than 8 years that might be related to the particular mutation (G24R) that was found in this patient that has not been described before.

In four of the five patients who were treated with lenalidomide, a significant reduction of the total clone size was observed.

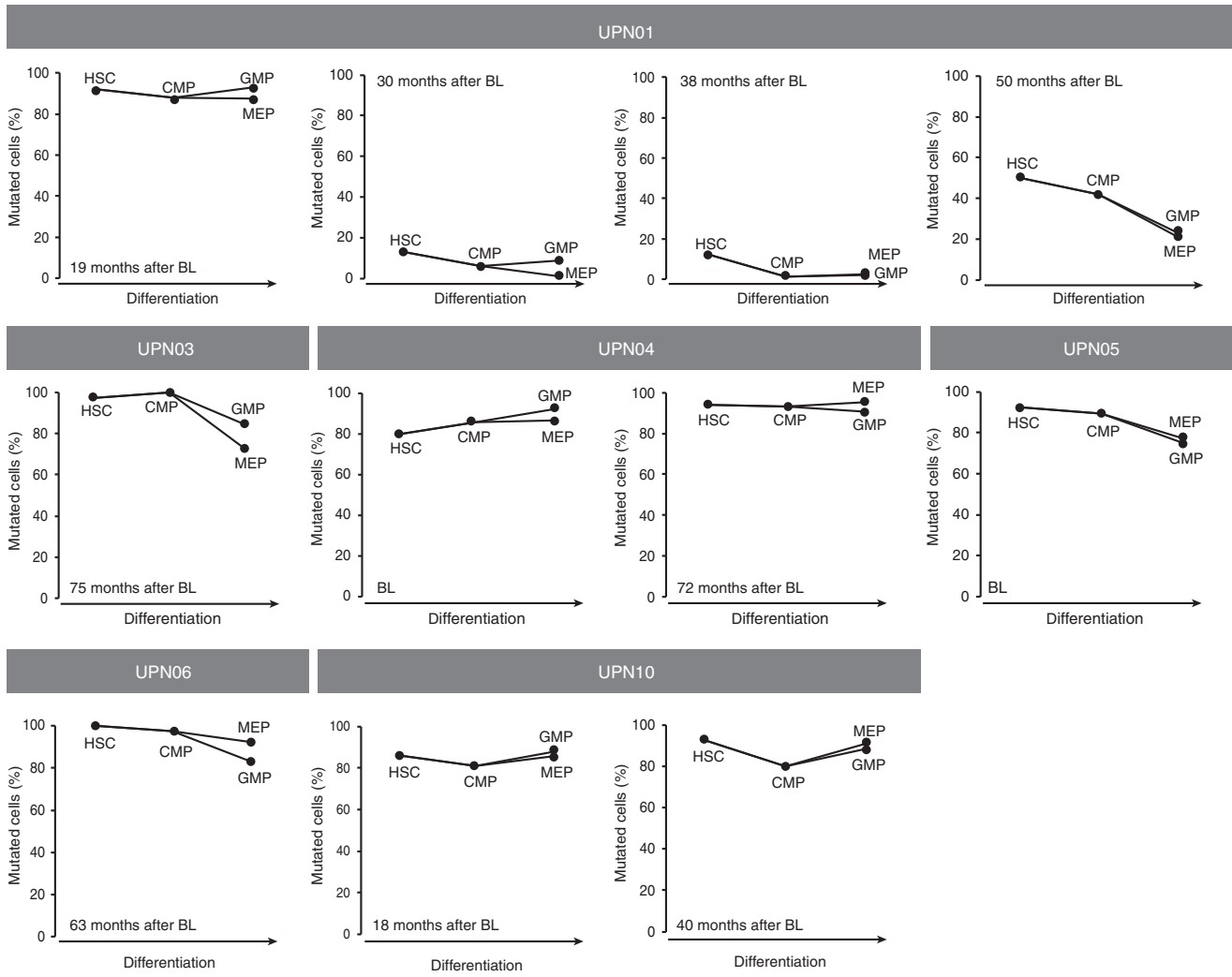

**Figure 5 | Percentage of MDS cells in various bone marrow stem and progenitor cell fractions.** From six MDS patients with sufficient material (UPN01, 03, 04, 05, 06 and 10), we sorted different bone marrow stem and progenitor cell fractions at various time points. Some minor differences in tumour burden are observed between the various fractions. BL, baseline; HSC, haematopoietic stem cell; CMP, common myeloid progenitor; GMP, granulocyte–macrophage progenitor; MEP, megakaryocyte–erythroid progenitor.

Interestingly, in three of the responding patients, preexisting, small clonal populations harbouring acquired mutations not shared with the MDS cells grew out upon the reduction of the number of MDS cells. In these patients, the application of disease-reducing treatment may have created an evolutionary bottleneck, after which repopulation may have occurred by a limited number of HSCs harbouring preexisting mutations. Similar observations have recently been described after induction chemotherapy in AML[27]. The data suggest that several scenarios may occur. Upon therapeutic reduction of the MDS clone a pattern resembling clonal haematopoiesis of indetermined potential may be observed, with clonal expansion of cells that do not carry any known driver mutation (like in UPN08)[28–30]. Furthermore, the reduction of the original MDS clone may create space for the outgrowth of preexisting cells that carry well-known driver mutations. This may lead to growth advantage during recolonization of the bone marrow after therapy, like in patient UPN10, in whom a *JAK2*-mutated clone expanded that did not progress beyond a clone size of 20% and did not undergo further genetic evolution. Finally, more proliferative and genetically instable clones may grow out (like in patient UPN09) that still may be derived from the initial MDS clone, but in which the early

common mutation was missed. Alternatively, these cells may represent a second *de novo* MDS.

After 4.5 years of treatment, UPN10 gradually lost the response to lenalidomide and underwent an allogeneic stem cell transplantation. At 19 months after transplantation, one of the del(5q) clones expanded, along with a clinical relapse. Interestingly, this clone was genetically identical to one of the clones that originally responded very well to lenalidomide. Therefore, the relapsing clone might have been lenalidomide sensitive, and restarting treatment might have been a valid option.

Two patients (UPN02 and UPN10) were treated with 5-azacitidine. In UPN02, the major clone decreased under 5-azacitidine treatment, whereas a subclone carrying an *EZH2* mutation expanded. After 5-azacitidine treatment was stopped, the *EZH2*-mutated subclone diminished and became undetectable, indicating that the *EZH2*-mutated subclone had a growth advantage and the major clone was diminished under 5-azacitidine treatment. UPN10 showed an improvement of haemogloblin levels and a reduction in clone size upon 5-azacitidine treatment. After 8 cycles the patient refused further treatment due to her poor condition. After discontinuation of 5-azacitidine treatment, the MDS clone re-expanded. This

observation is in contrast with the recently published data by Merlevede et al.[31], in which no decrease in clone size was observed in monocytes from chronic myelomonocytic leukaemia patients treated with hypomethylating agents.

The analysis of mutational burdens in various stem and progenitor fractions indicates that in general, no gross differences were observed between the different cell populations. This suggests that both early and late MDS-associated mutations originate in HSCs that are still capable of differentiation into the various myeloid lineages, in line with the analysis of stem cell fractions reported by Woll et al.[13] In addition, the mutational burdens in BM and PB were quite comparable. This suggests that the more patient-friendly monitoring of patients on the basis of peripheral blood is probably accurate[32], comparable with the monitoring of BCR-ABL levels in peripheral blood of chronic myeloid leukaemia patients[33].

Our study shows that various clonal evolution patterns can be observed in MDS patients treated with and without disease-modifying therapy. Monitoring of the genetic landscape during the disease may help to guide treatment decisions.

## Methods

**Patient samples.** Eleven MDS patients (7 males and 4 females) were selected based on having a long disease course (2.5–11 years of follow-up, median 7) and many sampling moments (5–31, median 7) (Table 1). Two categories of patients were analysed: patients who received supportive treatment only ($n = 6$) and patients who were treated with lenalidomide ($n = 5$). Two patients of the latter group also received 5-azacitidine. BM and PB from these patients were obtained at multiple time points. The study was conducted in accordance with the Declaration of Helsinki and institutional guidelines and regulations from the Radboudumc Nijmegen (IRB number: CMO 2013/064), and included informed consent by all patients. The patient characteristics are listed in Table 1. Morphology of BM cells was examined using standard May-Grünwald-Giemsa stainings.

**DNA isolation and amplification.** DNA was isolated from PB or BM of MDS patients using the NucleoSpin Blood QuickPure kit (Macherey Nagel, Düren, Germany) according to the manufacturer's protocol. In addition, BM and PB mononuclear cells (MNCs) and PB granulocytes were obtained after Ficoll-1077 density gradient separation. BM or PB cells were slowly added on top of a layer with Ficoll-Paque PLUS (density 1.077) (GE Healthcare, Chicago, IL, USA). After centrifugation at $700\,g$ for 20 min, MNCs were present on top of the Ficoll layer and granulocytes (and red bloods) underneath. These two cell fractions were collected separately, after which DNA was isolated. When the extraction yield was insufficient ($<5\,\mu g$) as measured with the Qubit fluorometer Quant-iT dsDNA BR Assay Kit (Thermo Fisher Scientific, Waltham, MA, USA), 80 ng of DNA was amplified using the Qiagen REPLI-g kit (Qiagen, Venlo, The Netherlands) in 4 parallel reactions (20 ng per reaction), according to the manufacturer's protocol.

**Karyotype analysis.** Bone marrow samples were cultured for 24–48 h in RPMI-1640 medium (Life Technologies, Carlsbad, CA, USA) supplemented with 10% fetal calf serum and antibiotics. After hypotonic treatment with 0.075 M KCl and fixation in methanol/acetic acid (3:1), microscopic slide preparations were prepared. Chromosomes were G-banded using trypsin (Life Technologies) and Giemsa and at least 20 metaphases were analysed in case of a normal karyotype, and at least 10 in case of an abnormal karyotype. Karyotypes were described according to the standardized ISCN 2013 nomenclature system[34].

**Fluorescence in situ hybridization.** Standard cytogenetic cell preparations were used for FISH. FISH was performed using commercially available probe kits for LSI EGR1/D5S23D5S721, LSI IGH/MYC/CEP 8 and D13s319/13q34 FISH, according to the manufacturer's specifications (Abbott Molecular, Des Plaines, IL, USA). Fluorescent signals of at least 200 interphase nuclei were scored and interpreted by two independent investigators. The cutoff values for both gains and losses were determined by statistical evaluation of FISH results from control tissue. For each probe the mean + 3 s.d. of false positive nuclei was taken as the cutoff level.

**T-cell culture.** Pure T cells were obtained from each patient by in vitro expansion of T cells from PB (or BM). Monocytes were first depleted by adherence to tissue culture flasks. The remaining cells were cultured for 14 to 21 days in IMDM medium (Life Technologies) supplemented with 10% human serum (PAA Laboratories GmbH, Pasching, Austria), interleukin-2 ($100\,\text{IU}\,\text{ml}^{-1}$) and CD3/CD28-coated Dynabeads (Thermo Fisher). The purity of the T cells was measured by flow cytometric analysis using the CD3 surface marker. When the

purity of the T cells exceeded 95%, DNA was isolated using the NucleoSpin Blood QuickPure kit.

**Mesenchymal stromal cell culture.** MSC lines were generated from five subjects. Bone marrow MNCs were obtained by Ficoll-1077 density gradient separation. BM-MNCs were seeded at a density of 8 to $23 \times 10^4$ cells cm$^{-2}$ in α-MEM medium (Sigma-Aldrich, St Louis, MO, USA) supplemented with heparin ($3.5\,\text{IU}\,\text{ml}^{-1}$) and 5% platelet lysate. Platelet lysate was prepared by freeze-thawing of platelets ($>0.8 \times 10^9$ platelets per ml), followed by centrifugation at $4,700\,g$ and collection of the supernatant. At 7 days after seeding, the culture medium was refreshed. Subsequently, cells were passed when 80% confluency was reached. After 7 days of culture, all floating and dead cells were washed away and a layer with MSCs remained. MSCs were cultured for up to 5 passages.

**CFU-GEMM culture and sequencing of single colonies.** PB-MNCs or BM-MNCs were seeded in methylcellulose media (10,000–25,000 cells per ml for BM and 100,000–200,000 cells per ml for PB) containing stem cell factor, interleukin-3, granulocyte–macrophage colony-stimulating factor and erythropoietin (H4434; Stem Cell Technologies, Vancouver, Canada) and incubated for 14 days at 37 °C with 5% $CO_2$. Individual colonies were collected on day 14 and washed with phosphate-buffered saline in a 96-well plate. Cells were lysed by adding 30 µl lysis buffer (TE-buffer + 0.5% Igepal-CA630 + 0.6 µl proteinase K ($10\,\text{mg}\,\text{ml}^{-1}$)) followed by incubating at 56 °C for 120 min and at 90 °C for 30 min. Subsequently, 1 µl of the lysate was used for each PCR reaction. Targeted amplicon-based deep sequencing was performed as described below. To exclude the possibility of reporting the results of mixed colonies, only colonies in which mutations were detected with a VAF of $>40\%$ were reported as positive.

**Sorting of myeloid progenitors.** 1 ml viably frozen bone marrow MNCs were thawed in the presence of 100 µl DNAse I ($2\,\text{mg}\,\mu\text{l}^{-1}$) and incubated for 10 min in a solution of 1.6 ml fetal calf serum, 10 µl heparin ($5,000\,\text{U}\,\text{ml}^{-1}$) and 100 µl MgSO$_4$ (0.22 µM). Subsequently, the myeloid progenitor cells were sorted according to a protocol adapted from Pang et al.[35] The cells were washed and stained with CD34-APC (Beckman Coulter, Brea, CA, USA), CD38-PE-Cy7 (BioLegend, San Diego, CA, USA), CD123-PE (BioLegend) and CD45RA-PB (BioLegend) monoclonal antibodies. Cells were analysed and sorted using a FACS Aria SORP flow cytometer and DIVA software (Becton Dickinson, Franklin Lakes, NJ, USA). Viable cells were selected based on forward scatter and side scatter profiles, and doublets were discriminated using forward scatter area versus width and side scatter area versus width. The HSC population was defined as CD34$^+$CD38$^-$. Within the CD34$^+$CD38$^+$ fraction, the common myeloid progenitor cells (CD123$^+$CD45RA$^-$), the granulocyte-macrophage progenitor cells (CD123$^+$CD45RA$^+$) and the megakaryocyte-erythroid progenitor cells (CD123$^-$CD45RA$^-$) were selected. DNA isolation from these cell fractions, followed by DNA amplification, was carried out using the Qiagen REPLI-g single cell kit (Qiagen) according to the manufacturer's protocol.

**Whole-exome sequencing.** WES to an average depth of $110 \times$ was performed on sequential BM-MNC ($n = 43$) and PB-MNC samples ($n = 2$) taken at regular time intervals (2 to 8 samples per patient). For all patients, DNA isolated from cultured T cells was used as a constitutive reference to exclude germline variants. Mutations significantly higher in the tumour cells than the T cells were listed as high confidence mutation and taken along in our analysis. In both, UPN02 and UPN03 one mutation was clearly affecting the T cells (VAF 19% and 24% respectively, see Supplementary Data 1), but in both cases the VAF was significantly higher in the tumour sample. Furthermore, for five patients DNA was available from cultured MSCs and used as additional germline control to ensure that no variants acquired in multipotent HSCs (and therefore also affecting T cells[36]) were incorrectly marked as germline variants and excluded. No MDS-associated mutations were found in the T cells of these five patients (Supplementary Table 4), indicating that the T cells were not part of the malignant clone.

Exome capture was performed using SureSelect Human All Exon V5 (Agilent Technologies, Santa Clara, CA, USA). Enriched exome fragments were then subjected to massively parallel sequencing using the HiSeq 2500 platform (Illumina, San Diego, CA, USA). Sequence alignment and mutation calling were performed using our in-house pipelines, as previously described[37], with minor modifications. Candidate mutations with (1) Fisher's exact $P \leq 0.001$ and (2) a VAF in tumour samples $\geq 0.07$ (to reduce false positive mutation calls) were selected. These variants were further filtered by excluding (1) synonymous SNVs, (2) SNVs in genes whose structure is not correctly annotated (complete open reading frame information is not available) and (3) SNVs listed as SNPs in the 1000 Genomes Project database (Nov 2010 release), dbSNP131 or our in-house SNP database. High-density SNP arrays were performed on DNA extracted from BM cells at several time points, allowing to correct VAFs for local copy number variations.

**Targeted deep sequencing using gene panels.** For one patient we analysed 2 samples collected after allogeneic stem cell transplantation using SureSelect (Agilent)-based targeted-capture sequencing for 72 known MDS driver genes

(Supplementary Table 2). Mutation calling was performed as previously described[3], with minor modifications. Germline SNVs were removed using WES data of paired germline control samples. Finally, we selected only mutations considered to be definitely oncogenic[2]. In addition, we used a myeloid gene panel (Trusight, Illumina) (Supplementary Table 1) to screen for driver mutations in unrelated clones.

**Targeted amplicon-based deep sequencing.** The candidate somatic variants detected by WES were validated and quantified by amplicon-based deep sequencing on an Ion Torrent Personal Genome Machine (Thermo Fisher Scientific) at high depth (aim 10,000 × coverage). Using this approach, mutational burdens were measured in all available PB and BM samples for each patient (Supplementary Data 3). Fragments with lengths of ~200 base pairs were amplified in two consecutive PCR reactions, PCR1 and PCR2, both of which were performed using Q5 Hot Start High-Fidelity Master Mix (New England Biolabs, Ipswich, MA, USA) according to the manufacturer's protocol. In PCR1, the target fragments were amplified and tagged with common sequence (CS)-tags (designed by Fluidigm, South San Francisco, CA, USA). For this purpose, sequence-specific primers were designed to obtain PCR fragments of ~200 base pairs. CS-tags were attached to these primers (see Supplementary Fig. 17 for primer strategy and Supplementary Tables 5 and 6 and Supplementary Data 4 for primer sequences). Depending on the primer pair, the best of three optimized touchdown PCR protocols was used (see Supplementary Table 7). In PCR2, primers containing a CS-tag, a barcode and an adapter (see Supplementary Fig. 17 for primer strategy and Supplementary Tables 5 and 6, and Supplementary Data 4 for primer sequences), were used to label the PCR fragments with a sample-specific Ion Xpress barcode (designed by Thermo Fisher Scientific) and add the adapters required for emulsion PCR. The second PCR was performed twice, once with the A adapter attached to the forward primer and the truncated P1 (trP1) adapter to the reverse primer (PCR2-A) and vice versa (PCR2-B), making bidirectional sequencing possible. For the PCR protocol for PCR2 see Supplementary Table 8. Subsequently, PCR products were pooled and purified with Agencourt AMPure XP beads (Beckman-Coulter, Fullerton, CA, USA) to eliminate primer dimers. After purification, the purity of the pool (based on expected fragment size) was measured on the Agilent 2200 TapeStation (Agilent Technologies) using the high-sensitivity D1000 ScreenTape assay (Agilent). The purified pool was diluted to 3 pg μl$^{-1}$ and loaded onto the Ion OneTouch system (Thermo Fisher Scientific) for emulsion PCR using the Ion PGM Template OT2 200 kit (Thermo Fisher Scientific), followed by an enrichment for loaded Ion Sphere Particles (ISPs). The quality of the enriched ISPs was checked with the Ion Sphere Quality Control Kit (Thermo Fisher Scientific) on the Qubit Fluorometer (Thermo Fisher Scientific). Subsequently, the ISPs were loaded onto an Ion 314, 316 or 318 v2 Chip (Thermo Fisher Scientific) and sequenced using the Ion PGM Sequencing kit v2 (Thermo Fisher Scientific) on the Ion Torrent Personal Genome Machine system (Thermo Fisher Scientific). All steps were performed according to the manufacturer's protocols. The sequencing data were mapped to the GRCh37 (hg19) reference genome build and variants were called with the SeqNext module of the Sequence Pilot software, version 4.2.2 (JSI Medical Systems, Ettenheim, Germany). Besides the automatic calling of variants, all locations wherein variants were detected by WES were manually inspected. A mutation was marked as validated by targeted deep sequencing when detected in the tumour sample (which was also used for WES) with a higher VAF than in the germline sample (at least 5% difference). The median validation rate per patient was 66.7%. Most mutations that could not be validated were mutations detected by WES in an amplified DNA sample (mainly insertions or deletions of a C or G), or mutations in genes that have a highly identical family member (likely incorrect mapping of WES reads). To determine an optimal cutoff VAF to discriminate true mutations from sequencing noise, we determined the sensitivity and specificity of Ion Torrent targeted deep sequencing. When we analysed the presence of 8 different mutations in 10 healthy donors, a VAF cutoff of 0.2% resulted in a specificity of 100% (Supplementary Table 9). In addition, we made a dilution series of 3 different SNPs and observed that a VAF of 0.1% could still accurately be detected (Supplementary Table 10). Based on this, we used a cutoff of 0.2%, which means 20/10,000 reads should harbour the mutation. In addition, the mutated base had to be the second highest base at the investigated position. This ensures that also in a more difficult sequence context the mutation exceeds the sequencing noise. In addition, a *FLT3*-ITD mutation was detected using fragment length analysis.

**Microarray-based genomic profiling (SNP array).** Microarray-based genomic profiling was carried out using the CytoScan HD array platform (Affymetrix, Inc., Santa Clara, CA, USA). Hybridizations were performed according to the manufacturer's protocols. The data were analysed using the Chromosome Analysis Suite software package (Affymetrix), using the annotations of reference genome build GRCh37 (hg19). For a comprehensive analysis of the microarray-based genomic profiling data, we used a previously developed filtering pipeline. The interpretation was performed using criteria adapted from Simons et al.[38] and Schoumans et al.[39] First, all aberrations affecting segments larger than 5 Mb (resolution of conventional karyotyping), regardless of gene content, were denoted as true aberrations. In addition, all aberrations affecting segments smaller than 5 Mb that coincided with known cancer genes (http://cancer.sanger.ac.uk/cancergenome/projects/census/, date of accession November 2012) were included. Since paired control DNA was not used, alterations that coincided with established normal genomic variants were excluded. For this approach, we used the publicly available 'Database of Genomic Variants' (http://projects.tcag.ca/variation) and, in addition, in-house databases of copy number alterations (CNAs) detected in ~1,000 healthy individuals studied with the CytoScan HD platform. Regions of copy-neutral loss of heterozygosity, also known as acquired uniparental disomy, were only considered if they were >10 Mb in size and if they extended towards the telomeres of the involved chromosomes, as reported by Heinrichs et al.[40] Finally, focal CNAs in the immunoglobulin and T-cell receptor genes were excluded from this study, as these CNAs generally represent the rearranged T-cell receptor and immunoglobulin genes present in the PB lymphocytes of the normal reference samples. All the data were also visually inspected to define alterations present in smaller proportions of cells and to eliminate alterations reported in regions with low probe density. Only aberrations fulfilling the above criteria were included in the genomic profiles and were described according to the standardized ISCN 2013 nomenclature system[34].

**Reconstructing clonal composition and evolution patterns.** Various software tools were tested to analyse clonal composition and evolution. However, different programs yielded different results, and close manual inspection showed imperfections in the patterns generated by all tested programs. Therefore, we constructed the clonal evolution patterns based on VAFs of all detected mutations at all time points, and included information from karyotyping, FISH and SNP arrays. For clonal reconstruction, all variants detected with a VAF of ≥0.2% were considered. Mutations were clustered based on the VAFs (corrected for ploidy) from all sequenced samples (PB and BM) at all different time points. The sequential order of mutational events and the most probable clonal evolution pattern were derived from these mutation clusters and their behaviour in time.

In UPN05, the clonal evolution pattern was calculated for the mononuclear myeloid cell fraction, rather than for the total BM-MNC fraction, as this patient developed bone marrow fibrosis and PB lymphocytosis, resulting in noncomparable sampling before and during treatment with romiplostim. In all other patients, lymphocyte counts were stable over time.

**Data availability.** Sequencing data (fastq files) of all 11 patients have been deposited into the NCBI Sequence Read Archive under accession number SRP094064. All other remaining data are available within the Article and Supplementary Files, or available from the authors on request.

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

## Acknowledgements

This work was supported by grants from ERA-NET JCT 2012 (TRIAGE-MDS), HOR-IZON2020 MDS-RIGHT and a grant from the Portuguese Foundation for Science and Technology (SFRH/BD/60391/2009), Grant-in-Aids from the Ministry of Health, Labor and Welfare of Japan, the Japanese Agency for Medical Research and Development (Health and Labour Sciences Research Expenses for Commission and Applied Research for Innovative Treatment of Cancer, the Project for Cancer Research And Therapeutic Evolution (P-CREATE)) and Japanese Society for the Promotion of Science (JSPS) KAKENHI (26221308, 15H05909, 26890016).

## Author contributions

P.d.S.-C., L.I.K., K.Y., B.A.v.d.R., S.O. and J.H.J. designed the study. T.d.W., N.M.A.B., P.M., G.H. and J.C. provided patient material and clinical data, and discussed progress. M.J.S.-K. performed and analysed the SNP arrays. K.Y., Y.S., K.C., H.T. and S.M. performed WES analysis. P.d.S.-C., L.I.K., T.N.K.-S., L.T.v.d.L., M.M. and A.O.d.G. performed deep sequencing and reconstruction of clonal evolution. R.K. performed CFU-GEMM cultures, and S.S. and M.D. performed bioinformatic analyses. J.H.J. and L.I.K. wrote the paper. All authors discussed the results and commented on the manuscript.

## Additional information

**Competing interests:** The authors declare no competing financial interests.

