## [Peer Review File · Nature Communications]

Reviewers' Comments:

Reviewer #1 (Remarks to the Author)

The authors have systematically sequenced and characterized the pattern of clonal evolution in 11 patients with MDS, half receiving disease-modifying therapy. Comments:

1. The patient cohort is billed as 'lower risk MDS.' UPN001 was classified as high risk by IPSS-R, but appears to have had a relatively benign clinical course. Certainly these outliers occur, but it is worth double-checking the IPSS-R parameters for this patient at time of first sampling to ensure there was no misclassification.
2. How were genes classified as 'MDS-associated' (Fig 1)? Does this overlap entirely with the panel used for targeted sequencing (ST5)?
3. The sequencing strategy and bioinformatic analysis requires some clarification. Were all the mutations depicted in Fig 1 and used to construct the clonal evolution profiles in Fig 2-3 detected by WES and validated by amplicon-based NGS? At the WES stage, what comparison was evaluated by the Fishers test? Why were synonymous mutations excluded (since they would still provide informative markers to depict clonal trajectories)? Was the VAF threshold of 7% arbitrarily selected, or can this be defended by empirical evidence? At the amplicon NGS stage, what criteria were used to conclude that a mutation was validated? What was the overall validation rate? What is their sensitivity of detection?
4. How were mutations clustered to infer clonal composition and evolution? Was the approach ad hoc, or based on a computational tool (e.g., Pyclone, Sciclone, etc)?
5. The pattern of clonal evolution observed in the 5 patients treated with disease-modifying therapy is fascinating. In 3 of these, clones emerge that are interpreted to be independent of the ancestral clone. This is a surprising result, but the evidence provided is not conclusive. In part, this relates to the previous comment on the clustering approach. However, more definitive evidence at the single cell level (e.g., FISH, ddPCR, genotyping) is required to conclusively demonstrate that these clones do not harbor the ancestral mutations. Moreover, in 2 of 3 cases, no recurrent genetic lesion was identified in the new expanding clones. This important observation leaves open another critical question: what kind of hematopoiesis is going on in these clones? Is this normal, polyclonal hematopoiesis that is recovering under the selective pressure of chemotherapy? Is this clonally-skewed, non-malignant hematopoiesis (ie, CHIP)? Is this a new, independent malignant process (e.g., therapy-related)? Or (most likely, in my opinion), is this an MDS clone, apparently not derived from the ancestral clone, in which the biologically-relevant driver mutation was not identified?
6. The use of cultured T cells as a surrogate for non-tumor tissue is problematic. There are many examples in the literature clearly demonstrating that T cells can be part of the MDS clone, creating the problem of false negatives when they are used to "subtract" mutation calls from the tumor population. The data provided by the authors (ST4) is not reassuring, since mutations were identified in T cells with VAFs (ranging from 7-11%), consistent with somatic acquisition. Although the mutations are not in recognizable MDS drivers, they demonstrate that the T cells are clonally-skewed and, therefore, likely harbor driver mutations that were missed.

Reviewer #2 (Remarks to the Author)

In the manuscript by da Silva-Coelho et. al., titled "Clonal Evolution in Lower Risk Myelodysplastic Syndromes", the authors report serial sequencing results from 2.5-11 years of follow-up from 11

MDS patients treated with supportive care (n=6) or lenalidomide (n=5). The authors document the patterns of clonal evolution in these 11 patients. Novel findings include monitoring clonal evolution in patients treated with supportive care and identification of potential emerging non-MDS clones in patients during lenalidomide treatment. The small sample size is limiting, but the approach is rigorous.

Major:

1. The targeted deep sequencing validation assay uses 2 consecutive rounds of PCR. The barcode is introduced in the second round of PCR. How do the authors avoid call variants that are generated from PCR errors during the first round of PCR?
2. The authors state a VAF sensitivity of 0.2% was achieved using targeted deep sequencing. What is the maximum sensitivity of the targeted deep sequencing assay? What is the specificity of the assay?
3. Please annotate which samples had REPLI-g whole genome amplification. This could alter the VAFs.
4. Include the variant and reference read count data and VAF in supplementary table 1 for all mutations. Please also include which code each mutation resides in based on the clonal models presented in Figures 2 and 3.
5. All the variants at all time-points that were sequenced should be provided with the variant and reference read count data and VAF. This will help the reader interpret the presence and absence of mutations and clones at each time-point.
6. Was there a difference in VAFs between bulk BM cells and the various cell fractions reported for HSC, CMP, GMP, and MEPs (data from Figures 4 and Supplementary Figures 5 and 6)? Prior results from Woll et. al. (Cancer Cell. 2014 Jun 16;25(6):794-808) suggest the HSPC fractions may typically have higher VAFs compared to bulk cells.
7. UPN10 = Please confirm that the clone model presented in Figure 3d reflects the Supplementary Figure 2 data. It is not clear why the purple clone is considered a "non-MDS" related clone versus a subclone of the dark blue clone and/or the orange subclone. In addition, it appears the MLL2 mutant clone and the light blue clone represent the same clone, not 2 distinct clones. Please justify how the clonal model was generated for UPN10 based on the VAFs.

Minor:

1. The title and abstract say "lower" risk MDS. However, 5 patients are categorized as 'Int' and 1 as 'high' by IPSS-R. These 6 cases could be viewed as higher risk by some readers and you may consider modifying 'lower'.
2. Consider converting the patients to the columns and genes to rows in Figure 1c and then ordering the patients based on those sharing similar gene mutations. This would be typical of many sequencing studies published for MDS and would allow easier comparison between cohorts.
3. Is there a difference in the transition/transversion ratio at baseline compared to later time points? It would be especially interesting to compare these ratios in treated versus untreated patients.
4. In UPN05, UPN06, UPN07, and UPN11 RAS pathway member mutations were present at late or secondary AML timepoints. Using deep amplicon sequencing were these mutations detected at the earliest timepoints sequenced?
5. Line 387 states that SNVs whose gene structure is not correctly annotated were removed. Please explain.
6. Is there a correlation between branching evolution and development of secondary AML?
7. Consider commenting on the effect that growth factor support may have in clonal evolution in patients treated with supportive care as most of the patients in Figure 2 were treated with growth factors.

Reviewer #3 (Remarks to the Author)

In this study, da Silva-Coelho and colleagues have performed a comprehensive, in-depth analysis of clonal evolution in low-risk MDS patients receiving lenalidomide or not. They describe the

intraclonal dynamics based on WES and SNP-array data for each individual case and interpret findings in relation to the clinical course/given therapy.

Comments:

Overtime analysis of clonal evolution/intraclonal dynamics using NGS have been reported in MDS in larger patient series, by e.g. Mossner et al, Blood 2016 and Chesnais et al, Blood 2016. However, these studies are not discussed and the authors must stress the novel aspects provided by their study.

Was targeted amplicon-based deep sequencing performed on all mutations detected in all cases? Were all samples for the respective patient analysed using deep-sequencing?

What was the VAF cutoff level for targeted amplicon-based deep sequencing? VAF percentages should be added for all variants in Supplemental Table 1.

The outline of the discussion should follow the presentation in the Results section.

Avoid speculative statements in the Discussion, for instance:

"In view of the observed evolutionary pattern, it would be very interesting to know what would have happened if lenalidomide treatment had been given intermittently after achieving complete remission. Possibly the TP53-mutated clone would not have been selected, preserving lenalidomide-sensitivity over a longer period of time."

"Our analysis indicates that continuation of the treatment might have been justified."

Response to Reviewer 1:

1. The patient cohort is billed as ‘lower risk MDS.’ UPN001 was classified as high risk by IPSS-R, but appears to have had a relatively benign clinical course. Certainly these outliers occur, but it is worth double-checking the IPSS-R parameters for this patient at time of first sampling to ensure there was no misclassification.

We checked the classification of the patients, and can confirm that they were correctly represented. However, the question of the reviewer implicitly also raises the following: the patients we included were selected *a priori* for the presence of multiple samples over a prolonged period of time. In doing so, most of them automatically fall in the lower risk categories. The title of our manuscript however could suggest that we included only lower risk category patients, whereas in fact, we included long term survivors. In practice this is quite similar, but not exactly the same; some higher risk patients indeed do survive for prolonged periods of time, as the reviewer correctly points out. To avoid the (unintended) suggestion that our analysis concerns only low-risk patients by the current classification scores, we removed the term ‘lower-risk’ from the title. (See also the minor comment #1 of reviewer 2)

2. How were genes classified as ‘MDS-associated’ (Fig 1)? Does this overlap entirely with the panel used for targeted sequencing (ST5)?

Genes that are described in the literature to be recurrently affected in myeloid malignancies are marked as MDS-associated (Fig 1). We added the literature references on which this is based to the legend of Fig 1. The panel used for targeted sequencing contained several additional genes (see supplemental Table 2), but none of these were found to be mutated.

During revision of the text, we realized that we referred to “genes implicated in MDS”, but a better definition is in fact “ genes implicated in myeloid malignancies”. We changed this in the legend of Fig 1 and in the text on page 3.

3. The sequencing strategy and bioinformatic analysis requires some clarification. Were all the mutations depicted in Fig 1 and used to construct the clonal evolution profiles in Fig 2-3 detected by WES and validated by amplicon-based NGS? At the WES stage, what comparison was evaluated by the Fishers test? Why were synonymous mutations excluded (since they would still provide informative markers to depict clonal trajectories)? Was the VAF threshold of 7% arbitrarily selected, or can this be defended by empirical evidence? At the amplicon NGS stage, what criteria were used to conclude that a mutation was validated? What was the overall validation rate? What is their sensitivity of detection?

Were all mutations detected by WES and validated by amplicon based NGS?: Indeed, all mutations that were detected by WES which could be confirmed by amplicon-based NGS were used for the construction of the clonal evolution patterns. FLT3-ITD mutations (which are known to be easily missed by WES) were analyzed separately using fragment length analysis. FLT3-ITD and one additional SRSF2 mutation that was not found by WES but by amplicon-based NGS (Trusight panel) were also taken along in the clonal reconstruction. To clarify that the variants detected by WES were all validated and quantified with targeted deep sequencing we included the following sentence on page 3; ‘Subsequently, all mutations were validated and quantified by targeted deep sequencing in all available samples of each patient (on average 10,616 fold coverage)’.

WES pipeline: We compared the counts of sequence reads supporting mutant and wild-type alleles for each mutation, and calculated the p_value using the Fisher’s test. Synonymous mutations could indeed be informative for the reconstruction of clonal evolution patterns as this reviewer points out, but this would require that these would also all have to be individually validated using specifically designed amplicon-based sequencing tests. Given the amount of information which could be derived from the non-synonymous mutations (which have a higher likelihood to be involved in the pathogenesis of the disease), we decided to base our clonal patterns on these alterations. The VAF

threshold of 7% for WES was previously defined (*Yoshida et al. Nature 2011*) to reduce false positive mutation calls. We incorporated this information in the materials and methods section.

Amplicon-based targeted deep sequencing: A mutation was marked as validated by targeted deep sequencing when detected in the tumor sample (which was also used for WES) with at least a 5% difference in VAF compared to the germline control. The median validation rate per patient was 66.7%. Most mutations which could not be validated were mutations detected by WES in an amplified DNA sample (mainly insertions or deletions of a C or G), or mutations in genes which have a highly identical family member (likely incorrect mapping of WES reads). We added this information to the supplementary methods section. To determine the specificity and sensitivity, we analyzed the presence of 8 different mutations in 10 healthy donors. In this analysis, a VAF cut-off of 0.2% resulted in a specificity of 100%. In addition, we made a dilution series of 3 different SNPs and observed that a VAF of 0.1% could still accurately be detected. Based on this, we used a cut-off of 0.2%. We added this information to the supplementary methods section.

4. How were mutations clustered to infer clonal composition and evolution? Was the approach ad hoc, or based on a computational tool (e.g., Pyclone, Sciclone, etc)?

Various software tools, including Sciclone and PhyloWGS were tested to analyze clonal composition and evolution. However, different programs yielded different results, and close manual inspection showed imperfections and evidently impossible details in the patterns generated by all programs. In addition, most of the available programs cannot not all accommodate karyotyping and FISH information. Even though the clonal evolution patterns generated by these programs grossly resembled the manually constructed patterns, we concluded that the available software is not yet 100% reliable for the analysis of samples like ours. In our view, this is partly due to the lack of sufficient experimental data that can be used to test and optimize the algorithms used in these programs. We expect that better programs will become available, but for the paper, we decided to rely on manual reconstruction of clonal patterns for all patients. We were reassured of our conclusions by the extra experiments we performed in response to point 5 (see below), in which mutational analysis and reconstruction of clonal evolution was based on individual colonies derived from single cells. These experiments confirm the correctness of our manual analysis. We added this information to the methods section.

5. The pattern of clonal evolution observed in the 5 patients treated with disease-modifying therapy is fascinating. In 3 of these, clones emerge that are interpreted to be independent of the ancestral clone. This is a surprising result, but the evidence provided is not conclusive. In part, this relates to the previous comment on the clustering approach. However, more definitive evidence at the single cell level (e.g., FISH, ddPCR, genotyping) is required to conclusively demonstrate that these clones do not harbor the ancestral mutations. Moreover, in 2 of 3 cases, no recurrent genetic lesion was identified in the new expanding clones. This important observation leaves open another critical question: what kind of hematopoiesis is going on in these clones? Is this normal, polyclonal hematopoiesis that is recovering under the selective pressure of chemotherapy? Is this clonally-skewed, non-malignant hematopoiesis (ie, CHIP)? Is this a new, independent malignant process (e.g., therapy-related)? Or (most likely, in my opinion), is this an MDS clone, apparently not derived from the ancestral clone, in which the biologically-relevant driver mutation was not identified?

In order to strengthen our conclusions, we performed additional experiments. To provide more evidence at the single cell level, we grew single colonies (CFU-GEMM) for the three mentioned patients harboring an independent clone. For all three patients, sequencing of these single cell-derived colonies proved that the mutations from the ancestral MDS clone were not present in cells belonging to the unrelated clone (see Fig 4), thereby confirming the clonal evolution pictures as depicted in Fig 3.

In addition, to determine whether we missed driver mutations in the unrelated clones, we ran a TruSight myeloid gene panel (Supplementary table 1) on one sample of each patient (UPN08,

UPN09, UPN10). In UPN08 and UPN10 no additional mutations were found. In UPN09 we found a SRSF2 mutation, which had been missed by whole exome sequencing (mutation around the VAF threshold of 7% used in WES). We added this mutation to all graphs and tables. The data suggest that several scenarios may occur: in patient UPN08, upon therapeutic reduction of the MDS clone, a pattern resembling clonal hematopoiesis of indetermined potential (CHIP) was observed. In patient UPN10, a JAK2 mutated clone expanded that did not progress beyond a clone size of 20% and did not undergo further genetic evolution. Finally, in patient UPN09 a more proliferative clone grew out that also acquired additional aberrations, including a well-known MDS-related mutation (SRSF2). Even though we performed whole exome sequencing at multiple time points, we cannot exclude that we still missed driver mutations. We added this to the manuscript (discussion section).

6. The use of cultured T cells as a surrogate for non-tumor tissue is problematic. There are many examples in the literature clearly demonstrating that T cells can be part of the MDS clone, creating the problem of false negatives when they are used to “subtract” mutation calls from the tumor population. The data provided by the authors (ST4) is not reassuring, since mutations were identified in T cells with VAFs (ranging from 7-11%), consistent with somatic acquisition. Although the mutations are not in recognizable MDS drivers, they demonstrate that the T cells are clonally-skewed and, therefore, likely harbor driver mutations that were missed.

We are aware of the fact that T-cells can be part of the malignant clone. Therefore we did the following:

- 1) In our pipeline mutations affecting T cells are not immediately excluded. Mutations significantly higher in the tumor cells than the T cells were listed as high confidence mutations and taken along in our analysis. In both, UPN02 and UPN03 one mutation was clearly affecting the T cells (VAF 19% and 24% respectively), but in both cases the VAF was significantly higher in the tumor sample. To make this more clear we marked these two mutations in Supplementary Table 3 and added this information to the materials and methods section.
- 2) As mentioned in the materials & methods section: For 5 patients DNA was available from cultured MSCs and used as additional germline control to ensure that no variants acquired in multipotent HSCs (and therefore also affecting T cells) were incorrectly marked as germline variants and excluded. No MDS-associated mutations were found in the T cells of these five patients (Supplementary Table 6), indicating that we did not exclude any important driver mutations. We added this information also to the main text on page 3.

We did detect mutations in the T cells, indicating that the T cells are indeed clonally-skewed, but no driver mutations were present in the T cells of all 5 tested patients when MSCs are used as germline control. The clonality seen in the T cells may be due to the culturing procedure used to obtain pure T cell populations.

Response to Reviewer 2:

Major:

1. The targeted deep sequencing validation assay uses 2 consecutive rounds of PCR. The barcode is introduced in the second round of PCR. How do the authors avoid call variants that are generated from PCR errors during the first round of PCR?

Based on our WES results we knew which mutations we had to look for during deep sequencing. We did not interpret all other called variants (among which variants introduced during PCR). Of course there is still a possibility that PCR errors occurred at the exact same position as we were interested in. We therefore set a threshold at a VAF of 0.2%, since some A, C, G and T's can be detected at every position. In addition, the mutated base was required to be the second highest base at the investigated position, ensuring that also in a more difficult sequence context the mutation exceeds the sequencing noise. We added this information to the supplementary methods section.

2. The authors state a VAF sensitivity of 0.2% was achieved using targeted deep sequencing. What is the maximum sensitivity of the targeted deep sequencing assay? What is the specificity of the assay?

We sequenced with a depth of 10,000x, and observed some noise at various positions. When we analyzed the presence of 8 different mutations in 10 healthy donors, a VAF cut-off of 0.2% resulted in a specificity of 100%. In addition, we made a dilution series of 3 different SNPs and observed that a VAF of 0.1% could still accurately be detected. These two tests made us decide to use a cut-off of 0.2%, which means around 20/10,000 reads should harbor the mutation. We added this information to the supplementary methods section and Supplementary tables 10 and 11 (see also the comment #3 in response to reviewer 1)

3. Please annotate which samples had REPLI-g whole genome amplification. This could alter the VAFs.

We now indicated in Supplementary Figure 3 for which time points REPLI-g amplified DNA was used for targeted deep sequencing (which was used in only a minority of the cases).

4. Include the variant and reference read count data and VAF in supplementary table 1 for all mutations. Please also include which code each mutation resides in based on the clonal models presented in Figures 2 and 3.

As requested, we made a separate Supplementary document (Supplementary document 2) which contains per sample and per mutation the reference read count, variant read count, total number of reads and the VAF. In addition, we added an extra column to Supplementary Table 3 to indicate to which clone a particular mutation belongs.

5. All the variants at all time-points that were sequenced should be provided with the variant and reference read count data and VAF. This will help the reader interpret the presence and absence of mutations and clones at each time-point.

See point 4.

6. Was there a difference in VAFs between bulk BM cells and the various cell fractions reported for HSC, CMP, GMP, and MEPs (data from Figures 4 and Supplementary Figures 5 and 6)? Prior results from Woll et. al. (Cancer Cell. 2014 Jun 16;25(6):794-808) suggest the HSPC fractions may typically have higher VAFs compared to bulk cells.

We indeed observed that in some samples a lower VAF was observed in bulk cells compared to sorted fractions, whereas in other samples the VAF of the bulk was more or less comparable to the sorted fractions, see examples below. One of the reasons why the VAFs are lower in bulk cells is likely that in the bulk sample also lymphoid cells are present which in most cases do not harbor these mutations and consequently lead to a decrease in the total VAF.

7. UPN10 = Please confirm that the clone model presented in Figure 3d reflects the Supplementary Figure 2 data. It is not clear why the purple clone is considered a “non-MDS” related clone versus a subclone of the dark blue clone and/or the orange subclone. In addition, it appears the MLL2 mutant clone and the light blue clone represent the same clone, not 2 distinct clones. Please justify how the clonal model was generated for UPN10 based on the VAFs.

For UPN10, we had the following reasons why we think the JAK2 clone is a separate unrelated clone:

- At time point three JAK2 does not fit in any of the other clones, see fig 3.
- JAK2 follows a completely other pattern (based on VAFs) than all other mutations, see supplementary Fig 3.
- The founding clone contains a CALR mutation. JAK2 and CALR mutations both affect the JAK-STAT pathway and therefore occur in a mutually exclusive manner. This makes it very unlikely that the JAK2 clone is derived from one of the other clones (all harboring the CALR mutation).
- The JAK2 clone increases when the del5q-containing clones (red and dark green clones) decrease under treatment. This is in line with what we see in UPN08 and UPN09.

We had the following reasons why MLL2 is a separate clone:

- In the last time point (2 different samples: PB-MNC and BM-MNC) MLL2 and LRRC34 are detectable with a VAF around 0.30-0.40%, whereas mutations specific for the light blue clone are absent.
- The fact that LRRC34 and MLL2 were completely undetectable in the samples just after transplantation, shows that it is not a difficult sequence context in which the mutated base can always be found in 0.30-0.40% of the reads.

These two points together strongly suggest that MLL2 and LRRC34 form a separate clone.

To provide more evidence on a single cell level, we performed extra experiments (also in response to reviewer 1), and grew single colonies (CFU-GEMM) for patient UPN10. Sequencing of these single cell-derived colonies showed that cells harboring a JAK2 mutation did not harbor mutations present in the other subclones, thereby proving that the JAK2-mutated clone is a separate unrelated clone, see the novel figure 4. In addition, single cell sequencing confirmed that cells harboring MLL2 and LRRC34 mutations (light green clone) did not harbor mutations in OCA2, FRMD8 and PRPS1L1 (light blue clone) and vice versa, see novel figure 4.

Minor:

1. The title and abstract say “lower” risk MDS. However, 5 patients are categorized as ‘Int’ and 1 as ‘high’ by IPSS-R. These 6 cases could be viewed as higher risk by some readers and you may consider modifying ‘lower’.

A similar comment was raised by reviewer 1 (comment #1): We agree that this might be misleading, therefore we deleted the words ‘lower risk’ in the title and in the text.

2. Consider converting the patients to the columns and genes to rows in Figure 1c and then ordering the patients based on those sharing similar gene mutations. This would be typical of many sequencing studies published for MDS and would allow easier comparison between cohorts.

As suggested we ordered the patients and genes, in this way it is easier for the reader to interpret which mutations are present in each patient, see Fig 1C.

3. Is there a difference in the transition/transversion ratio at baseline compared to later time points? It would be especially interesting to compare these ratios in treated versus untreated patients.

As suggested we performed the analysis by comparing the type of mutations at baseline (mutations detected with a VAF >0.2%) with mutations that occur later in the disease course. In addition we compared the supportive care vs the lenalidomide group, see Supplementary Fig. 2.

4. In UPN05, UPN06, UPN07, and UPN11 RAS pathway member mutations were present at late or secondary AML timepoints. Using deep amplicon sequencing were these mutations detected at the earliest timepoints sequenced?

In the first sample of UPN05, UPN06 and UPN11 the RAS mutation was absent or detected in only a few reads below our threshold of 0.2%:

- UPN05: VAF 0.05%
- UPN06: VAF 0.08%
- UPN11: VAF 0%.

In UPN07 both NRAS (VAF 40%) and KRAS (VAF 12%) were detected in the first sample, as also depicted in the clonal evolution picture.

This information is now available for the reader in a separate Supplementary document (Supplementary document 2) which contains per sample and per mutation the reference read count, variant read count, total number of reads and the VAF.

5. Line 387 states that SNVs whose gene structure is not correctly annotated were removed. Please explain.

When a variant is located in a gene structure which is not correctly annotated (complete open reading frame information is not available) ANNOVAR reports unknown for exonic_variant_function; in other words, although the variant is clearly within an exon, we cannot say for sure how it affects protein sequence as the open reading frame annotation is incomplete. We removed those SNVs since we focused on elucidating (possible) driver mutations involved in clonal evolution and drug resistance. To clarify this, we added one sentence to the materials and methods section.

6. Is there a correlation between branching evolution and development of secondary AML?

1/5 patients with a linear clonal evolution (only looking at MDS clones, and not the MDS-unrelated clones) and 2/5 patients with a branching evolution developed AML. The number of patients in our study is however too small to draw conclusions regarding a possible correlation between branching evolution and development of secondary AML.

7. Consider commenting on the effect that growth factor support may have in clonal evolution in patients treated with supportive care as most of the patients in Figure 2 were treated with growth factors.

Growth factors are administered to patients to stimulate hematopoiesis. Specific mutations might make a cell more responsive to growth factors. Therefore it is possible that different subclones respond differently to growth factors resulting in clonal evolution. We added this comment to the discussion.

Response to Reviewer 3:

1. Overtime analysis of clonal evolution/intraclonal dynamics using NGS have been reported in MDS in larger patient series, by e.g. Mossner et al, Blood 2016 and Chesnais et al, Blood 2016. However, these studies are not discussed and the authors must stress the novel aspects provided by their study.

These studies appeared in 2016 and we agree that our text now should be updated. We now discuss these references in the introduction, and the discussion sections.

2. Was targeted amplicon-based deep sequencing performed on all mutations detected in all cases? Were all samples for the respective patient analysed using deep-sequencing?

Indeed, all mutations in all patient samples were analyzed with targeted deep sequencing. To clarify this we included the following sentence on page 3; ‘Subsequently, all mutations were validated and quantified by targeted deep sequencing in all available samples of each patient (on average 10,616 fold coverage)’.

3. What was the VAF cutoff level for targeted amplicon-based deep sequencing? VAF percentages should be added for all variants in Supplemental Table 1.

The VAF cut-off level for amplicon-based deep sequencing was 0.2% as mentioned in the materials and methods section (‘Reconstructing clonal composition and evolution patterns’). Why we choose this cut-off is explained in the supplementary methods section.

In addition, we made a separate Supplementary document (Supplementary document 2) which contains per sample and per mutation the reference read count, variant read count, total number of reads and the VAF.

4. The outline of the discussion should follow the presentation in the Results section.

We changed the outline of the discussion, now starting with the patients treated with supportive care followed by the patients treated with lenalidomide, comparable to the outline of the results section.

5. Avoid speculative statements in the Discussion, for instance:

“In view of the observed evolutionary pattern, it would be very interesting to know what would have happened if lenalidomide treatment had been given intermittently after achieving complete remission. Possibly the TP53-mutated clone would not have been selected, preserving lenalidomide-sensitivity over a longer period of time.”

“Our analysis indicates that continuation of the treatment might have been justified.”

We adapted the text to avoid too much suggestive statements that are not fully supported by experimental data. Where necessary we added that further research would be necessary to prove or disprove the indicated possibilities.

Reviewers' Comments:

Reviewer #1 (Remarks to the Author)

The authors have adequately addressed the previous criticisms. Additional (minor) comments:

1. In Figure 1, panel A: consider changing the legend from 'MDS-associated' to 'Myeloid Malignancy-associated' to conform to the changed nomenclature in the text.
2. In Figure 4, consider changing colors to match those used in Figure 3. The JAK2 clone in UPN010 is clearly independent of other clones, but it is hard to interpret clonal relationships for the other mutations.
3. Page 4, line 109: consider changing "...all samples," to "...all samples from a given patient." As written, this statement could be misinterpreted.
4. Reconsider use of the terminology: "non-MDS-related clonal populations." This is perhaps the most interesting finding in the paper and will be highly cited, so it is important to clearly communicate the authors' thinking. The authors have now convincingly demonstrated that in some patients, clones emerge that are not derived from the ancestral clones. The chosen terminology goes further and implies that they are not disease-related. Admittedly, they do not appear to be associated with major changes in clinical course, but do the authors think that these clones are not "MDS-related" (including the emerging SRSF2 mutated clone)? "Unrelated clone" or some other neutral terminology might be better. This could also use a slightly expanded discussion (page 9), where all scenarios are laid out (e.g., expansion of a CHIP clone, a new therapy-related MN clone, a second de novo MDS clone, etc). The authors state in their cover letter that a statement was included in the discussion acknowledging the possibility that the rising clones could be independent MDS-related clones in which the driver mutation was missed, but I cannot locate this language in the discussion.

Reviewer #2 (Remarks to the Author)

The authors have addressed my comments.

Reviewer #3 (Remarks to the Author)

The authors have performed a thorough revision of their manuscript and addressed this Reviewer's concerns satisfactorily. No further comments.

Response to Reviewer 1:

The authors have adequately addressed the previous criticisms. Additional (minor) comments:

1. In Figure 1, panel A: consider changing the legend from ‘MDS-associated’ to ‘Myeloid Malignancy-associated’ to conform to the changed nomenclature in the text.

Done as recommended.

2. In Figure 4, consider changing colors to match those used in Figure 3. The JAK2 clone in UPN010 is clearly independent of other clones, but it is hard to interpret clonal relationships for the other mutations.

As suggested we changed the colors in the figure, so that the color of each colony is matched to the color of the clone (fig 3) were it is derived from.

3. Page 4, line 109: consider changing “...all samples,” to “...all samples from a given patient.” As written, this statement could be misinterpreted.

Done as recommended.

4. Reconsider use of the terminology: “non-MDS-related clonal populations.” This is perhaps the most interesting finding in the paper and will be highly cited, so it is important to clearly communicate the authors’ thinking. The authors have now convincingly demonstrated that in some patients, clones emerge that are not derived from the ancestral clones. The chosen terminology goes further and implies that they are not disease-related. Admittedly, they do not appear to be associated with major changes in clinical course, but do the authors think that these clones are not “MDS-related” (including the emerging SRSF2 mutated clone)? “Unrelated clone” or some other neutral terminology might be better. This could also use a slightly expanded discussion (page 9), where all scenarios are laid out (e.g., expansion of a CHIP clone, a new therapy-related MN clone, a second de novo MDS clone, etc). The authors state in their cover letter that a statement was included in the discussion acknowledging the possibility that the rising clones could be independent MDS-related clones in which the driver mutation was missed, but I cannot locate this language in the discussion.

We agree with the reviewer and therefore changed ‘non-MDS-related clone’ into ‘unrelated clone’ throughout the text. In addition, we adapted the discussion so that all possible scenarios are more thoroughly explained. Here, we also acknowledged that the initial (driver) mutation might have been missed.